# Cellular Sensors and Viral Countermeasures: A Molecular Arms Race between Host and SARS-CoV-2

**DOI:** 10.3390/v15020352

**Published:** 2023-01-26

**Authors:** Haoran Sun, Jasper Fuk-Woo Chan, Shuofeng Yuan

**Affiliations:** 1Department of Infectious Disease and Microbiology, The University of Hong Kong-Shenzhen Hospital, Shenzhen 518009, China; 2State Key Laboratory of Emerging Infectious Diseases, Carol Yu Centre for Infection, Department of Microbiology, School of Clinical Medicine, Li Ka Shing Faculty of Medicine, The University of Hong Kong, Hong Kong, China; 3Centre for Virology, Vaccinology and Therapeutics, Hong Kong Science and Technology Park, Hong Kong, China

**Keywords:** SARS-CoV-2, innate immune response, pattern recognition receptors

## Abstract

Severe acute respiratory syndrome coronavirus 2 (SARS-CoV-2) is the causative agent of the coronavirus disease 2019 (COVID-19) pandemic that has caused disastrous effects on the society and human health globally. SARS-CoV-2 is a sarbecovirus in the *Coronaviridae* family with a positive-sense single-stranded RNA genome. It mainly replicates in the cytoplasm and viral components including RNAs and proteins can be sensed by pattern recognition receptors including toll-like receptors (TLRs), RIG-I-like receptors (RLRs), and NOD-like receptors (NLRs) that regulate the host innate and adaptive immune responses. On the other hand, the SARS-CoV-2 genome encodes multiple proteins that can antagonize the host immune response to facilitate viral replication. In this review, we discuss the current knowledge on host sensors and viral countermeasures against host innate immune response to provide insights on virus–host interactions and novel approaches to modulate host inflammation and antiviral responses.

## 1. Introduction

It has been over 3 years since the discovery of severe acute respiratory syndrome coronavirus 2 (SARS-CoV-2). Although multiple vaccines and antiviral compounds have been developed and used clinically, the coronavirus disease 2019 (COVID-19) pandemic continues to cause significant morbidity, mortality, and socioeconomic disruptions globally. SARS-CoV-2 belongs to the *Coronaviridae* family with an RNA genome of approximately 29.9Kb that undergoes continuous mutations and recombinations. Such mutations and recombinations could give rise to viruses with altered properties including replication and transmission efficiency, or with altered antigenicity that could evade from existing specific immunities elicited either by vaccination or infection [1]. There have been five variants of concern declared by the World Health Organization (WHO), namely, Alpha, Beta, Gamma, Delta, and Omicron. Currently, Omicron is the most predominant variant worldwide that has further evolved into novel subvariants with distinct antigenic properties capable of evading neutralizing antibody response elicited by past infection and/or vaccination [2,3].

Viruses are obligatory intracellular “parasites” [4]. After attachment and entry into susceptible cells, they rely mostly on replication machines of host cells to support viral replication and generation of progeny virions. On the other hand, host cells have immune responses against virus replication and are able to control the infection. Host immune responses could be divided into innate immune responses and adaptive immune responses [5]. Adaptive immune response is usually elicited after infection or vaccination, which requires a longer period and usually targets a specific antigen; on the other hand, the innate immune response could recognize viral infection and rapidly drive the innate cellular responses including the production of interferons (IFNs) and tumor necrosis factors (TNFs) to restrict viral infection [5].

The SARS-CoV-2 genome encodes 29 proteins, 4 of which were structural proteins, namely, Spike, Envelope, Membrane, and Nucleocapsid, together with 9 accessary proteins and 16 non-structural proteins (nsp1–nsp16) in polyprotein form and cleaved by viral proteases nsp3 (papain-like protease, PLpro) and nsp5 (3C-like cysteine protease, 3CLpro or Mpro) [6] (Figure 1). Upon viral attachment to host cells, the viruses fuse to the cellular membrane and release the viral genome to initiate the virus replication cycle. Innate immunity, the natural resistance of the host against pathogens, can eliminate pathogens through the production of IFNs, pro-inflammatory cytokines, and the formation of inflammasomes [7]. The initiation of innate immune response depends on the recognition of pathogen-associated molecular patterns (PAMPs) and danger-associated molecular patterns (DAMPs) by pattern recognition receptors (PRRs), including toll-like receptors (TLRs), retinoic-acid-inducible gene I (RIG-I)-like receptors (RLRs), NOD-like receptors (NLRs), C-type lectin receptors (CLECs), and absent in melanoma 2 (AIM2)-like receptors [8]. These receptors mediate pathogen recognition, and then lead to intracellular signaling and subsequently the synthesis of various cytokines. These cytokines then recruit other immune cells, regulate adaptive immune responses, and inhibit viral spreading. PRRs are found in different locations in the cells and cellular components in the blood and tissues to recognize viral PAMPs [9]. Recognition of PAMPs is the first and most important step to initiate the innate immune response, thus PRRs play an essential role in SARS-CoV-2 detection and restriction; therefore, in this review, we will focus on the role of PRRs involved in SARS-CoV-2 recognition, virus antagonism, and possible roles of these sensors in treating COVID-19 to relieve virus infection and inflammation.

## 2. SARS-CoV-2 and TLRs

Toll-like receptors (TLRs) were the first and best-characterized family of PRRs to be discovered, located on the surface of cells or inside cells, and recognize classes of pathogens including Gram-positive bacteria, Gram-negative bacteria, fungi, and DNA or RNA viruses [10]. Furthermore, TLRs are type I integral membrane proteins, and there are 12 members in mice and 10 members in humans (TLR1 to TLR10); TLR3, TLR7, TLR8, and TLR9 are localized primarily in endolysosomes, whereas the rest of the TLRs are expressed on the cell surface and their cellular location is tailored to enable them to respond optimally to the particular microbial ligands they recognize [11]. Structurally, TLRs are composed of an extracellular domain with leucine-rich repeats (LRRs) that mediate the recognition of PAMPs, a transmembrane domain, and a cytoplasmic Toll/IL-1 receptor (TIR) domain that initiates downstream signaling [12]. Upon activation, TLRs induce downstream signaling via recruiting key adaptors, MyD88 (myeloid differentiation factor 88) and TRIF (TIR domain-containing adaptor-inducing IFN-β factor) [13].

TLR2 is involved in sensing different kinds of pathogens including bacteria, viruses, fungi, and parasites [14]. By examining the activation of inflammatory signaling pathways, Zheng et al. showed that TLR2 sensed the SARS-CoV-2 envelope (E) protein as its ligand. E protein could directly bind human TLR2 and induce expression of TNF, IFN-γ, interleukin-6 (IL-6), and IL-1β in human peripheral blood mononuclear cells (PBMCs). In mouse models, administration of the E protein from SARS-CoV-2 also triggered the recruitment of large numbers of inflammatory cells in the lungs of WT mice but not in the lungs of TLR2-deficient mice, and elevated the expression level of IL-6, C-X-C motif chemokine ligand 10 (CXCL10), and granulocyte colony stimulating factor (G-CSF), thus contributing to the cytokine storm and lung lesions [15]. Since expression of human SARS-CoV-2 receptor angiotensin-converting enzyme 2 (ACE2) is limited in the neural central system, and TLR2 plays an important role in the pathogenesis of neurodegenerative diseases such as Alzheimer’s disease and Parkinson’s disease, there is a hypothesis that TLR2 may play a critical role in the response to SARS-CoV-2 infiltration in the CNS, thus resulting in the induction or acceleration of AD and PD pathologies in patients [16]. Another study by Khan et al. demonstrated that SARS-CoV-2 S protein could be sensed by TLR2 and activates the nuclear factor kappa-light-chain-enhancer of the activated B cells (NF-κB) pathway, leading to the expression of inflammatory mediators including IL-6, TNF-α, and IL-1β in innate immune and epithelial cells, and the sensing and recognition by TLR2 require dimerization with either TLR1 or TLR6 [17]. Additionally, excessive neutrophil extracellular traps correlated with SARS-CoV-2 infection severity contribute to immunothrombosis, thereby leading to extensive intravascular coagulopathy and multiple organ dysfunction [18]. TLR2 and CLEC5A (C-type lectin domain-containing 5A) are critical in activating platelets to produce extracellular vesicles, which further enhance thromboinflammation [19]. Consistent to a molecular docking study [20], a surface plasmon resonance (SPR) assay confirmed that trimeric SARS-CoV-2 S protein directly binds to TLR4 with a high affinity of ∼300 nM, which is comparative to many virus–receptor interactions, thus activating downstream cytokines and IFNs [21]. These results indicate that SARS-CoV-2 structural spike and envelope proteins could be sensed by TLR2 and TLR4, and induce inflammatory processes.

TLR3 is located within endosomes and is a receptor for double-stranded RNA (dsRNA). However, since many viruses have dsRNA within their genomes or generate dsRNA in their viral life cycles, TLR3 can sense the presence of single-stranded RNA (ssRNA), dsRNA, and DNA viruses [22]. Upon SARS-CoV-2 infection, TLR3 protein expression increases and might act via NF-κB and interferon-regulatory factor 3 (IRF3); however, no detailed mechanism studies have been reported [23]. Another study by Tripathi and colleagues found that TLR3 is involved in causing non-senescent cells to become senescent through TLR3, either vesicular stomatitis virus (VSV)-based pseudovirus or SARS-CoV-2 can amplify the tissue-destructive senescence-associated secretory phenotype (SASP) through TLR3, with senescent cells becoming more pro-inflammatory than non-senescent cells after viral exposure, thus contributing to SARS-CoV-2 morbidity [24]. Unlike TLR3, TLR7/8 senses ssRNA and then activates subsequent production of type I IFNs and inflammatory cytokines in a MyD88-dependent pathway [25,26,27]. TLR7 is expressed mainly on plasmacytoid dendritic cells (pDCs), which play an important role as the major source of type I IFN, while TLR8 is expressed mainly on myeloid cells such as monocytes/macrophages and myeloid DCs (mDCs) that induce pro-inflammatory cytokines and activates T cells [28,29,30]. Bioinformatic analysis has shown that compared with SARS-CoV, SARS-CoV-2 contains more ssRNA fragments that could be recognized by TLR7/8 and thus may contribute to a more robust proinflammatory response via TLR7/8 recognition and cause acute lung injury [31]. Salvi et al. confirmed that ssRNA of SARS-CoV-2 could directly activate endosomal TLR7/8 and MyD88, products of virus endosomal processing that potently activate the IFN and inflammatory responses downstream. In vivo experiments showed that ssRNA SARS-CoV-2-associated molecular patterns (SAMPs) induced MyD88-dependent lung inflammation characterized by accumulation of proinflammatory (TNF-α, IL-1β, IL-6, IFN-α, and IFN-γ) and cytotoxic (granzyme B and TRAIL) mediators and immune cell infiltration, as well as splenic DC phenotypical maturation [32].

Given the important role of TLRs in immune response, variants may undermine immune response and may affect infection outcomes. It has been reported that TLR3 absence confers increased survival with improved macrophage activity against pneumonia [33], and recently, a functional L412F polymorphism (rs3775291; c.1234 C > T) with decreased expression level on cell surfaces, is related to SARS-CoV-2 susceptibility and mortality due to poor recognition of viral dsRNAs [34,35]. Further investigation by Croci et al. showed that L412F inhibited autophagy, increased frequency of autoimmune disorders. In addition, co-morbidity was found in L412F COVID-19 males with specific class II HLA haplotypes prone to autoantigen presentation, indicating that TLR3 L412F is a severity marker in COVID-19 infection [36]. Additionally, inborn errors of TLR3- and IRF7-dependent type I IFN immunity can underlie life-threatening COVID-19 pneumonia in patients with no prior severe infection [37], further demonstrating the important role of TLRs in immune response and immune pathogenesis. Therefore, future investigations of key variants in TLRs are required for a better understanding of human genetic determinants of critical COVID-19 pneumonia.

## 3. SARS-CoV-2 and NLRs

The NLRs are a large family of cytosolic proteins that can be activated by intracellular PAMPs and substances that alert cells to damage or danger (DAMPs and other harmful substances). They play major roles in activating beneficial innate immune and inflammatory responses; however, some NLRs also trigger inflammation that causes extensive tissue damage and disease. The human genome encodes 23 NLR proteins divided into three major groups based largely on their domain structure: NLRCs (some of which have caspase recruitment domains, or CARDs), NLRBs (which have baculovirus inhibitory repeat (BIR) domains), and NLRPs (which have pyrin domains, or PYDs) [38]. Among them, activation of NLRP3 is closely related to sensing of a variety of DAMPs and PAMPs. NLRP3 activation can further induce the formation of inflammasome multiprotein complexes, activate the inflammatory caspases, and lead to the production of mature IL-1 and IL-18, and pyroptotic cell death [39].

The NLRP3 inflammasome consists of NLRP3, an adaptor protein (apoptosis-associated speck-like protein containing a caspase recruitment domain, ASC) and an effector protein (Caspase-1), which can be activated upon infection by a range of RNA viruses, while neither NLRP3 itself nor the other inflammasome components can bind RNA [39,40]. Activation of the NLRP3 inflammasome could rely on members of the DExD/H-box family of helicases to sense viral RNA; convergence of RNAse L pathway and DExD/H-box helicase, DHX33, and mitochondrial adaptor protein, MAVS (mitochondrial antiviral signaling protein) [41]; or Z-DNA-binding protein 1 (ZBP1)/DNA-dependent activator of IFN-regulatory factors (DAI) to sense influenza viral RNAs [42]. Increasing evidence has shown that NLRP3 inflammasome is activated in response to SARS-CoV-2 infection, is active in COVID-19 patients, and participates in the pathophysiology [43,44,45].

Previous studies focusing on SARS-CoV have confirmed that viroporins with ion channel activity, including E [46], ORF3a [47], and ORF8b [48] are involved in NLRP3 inflammasome activation. Upon SARS-CoV-2 infection, human macrophages activate inflammasomes, produce inflammatory cytokines, and drive pyroptosis that potentially contributes to the COVID-19 pathology [49]. Pan et al. showed that the 260aa to 340aa of the nucleocapsid protein (N) can directly bind NLRP3 and promote NLRP3 inflammasome activation by facilitating the interaction between NLRP3 with ASC to induce hyperinflammation. In the mouse model, NLRP3 has an important function in N-protein-induced lung injury as inflammatory factors including IL1β, IL-6, and the occurrence of lung injury was observed [50]. Upon inflammasome activation, pore-forming cell death executor Gasdermin D (GSDMD) could be cleaved by caspase 1 leading to a lytic inflammatory cell death, pyroptosis [51]. In monocytes, it has been proposed that SARS-CoV-2 N protein can directly bind the GSDMD linker region and further hinders GSDMD processing by caspase-1, which leads to lytic inflammatory cell death and pyroptosis [52]. Another study by DeDiego et al. demonstrated that in bone-marrow-derived macrophages (BMDMs), SARS-CoV-2 E protein suppresses the ER stress response and NLRP3 inflammasome activation, similar to SARS-CoV E [53]. In mice lung, priming and activation of the NLRP3 inflammasome activation by poly(I:C) to simulate the effects of viral RNA could also be reduced by SARS2 E protein, while in human and murine macrophages, E protein enhances NLRP3 inflammasome activation when stimulated with LPS and poly(I:C). These facts suggest that E protein suppresses NLRP3 inflammasome activation during the early stages of infection while in the later stages, it may enhance NLRP3 inflammasome activation [54]. Moreover, in mice and human microglia, SARS-CoV-2 spike could activate the NLRP3 inflammasome via the ACE2-NF-κB axis, suggesting that besides lung pathogenesis, SARS-CoV-2 also has a negative impact on neurodegenerative diseases [55].

Besides structural proteins, SARS-CoV-2 viroporin ORF3a was confirmed to prime and activate the NLRP3 inflammasome via both ASC-independent and ASC-dependent modes, with the involvement of K^+^ efflux, leading to the release of the proinflammatory cytokine IL-1β and causing epithelial cells death [56]. Sun et al. showed SARS-CoV-2 nsp6 protein can directly interact with ATPase H+ transporting accessory protein 1 (ATP6AP1), a vacuolar ATPase proton pump component, and inhibit its cleavage-mediated activation to impair lysosome acidification, thereby causing autophagic flux stagnation to instigate NLRP3 inflammasome activation and pyroptosis [57].

The role of NLRP3 in inflammasome during SARS-CoV-2 has been extensively studied and other NLRs have also been reported to be involved in sensing viral infections. Yin et al. demonstrated that NOD1 (NLRP1), a well-known sensor for bacterial peptidoglycans, could also sense SARS-CoV-2 infection in lung epithelial Calu-3 cells, and activation of NOD1 may lead to IFN production through the NF-κB pathway activation [58]. Previous studies have shown that NLRP1 could sense picornaviruses [59], and a recent study by Planès et al. identified that NLRP1 also senses SARS-CoV-2. They demonstrated that NLRP1 could be cleaved at the Q333 site by SARS-CoV-2 3CLpro and it triggers inflammasome assembly and cell death. However, SARS-CoV-2 can also counteract the inflammasome signaling by directly targeting and inactivating GSDMD downstream of NLRP1, as GSDMD could be cleaved by SARS-CoV-2 nsp5 at Q333 site. Alternatively, NLRP1 drives apoptotic caspase-3 activation and promotes pyroptotic Gasdermin-E-dependent pyroptosis, thus restricting the generation and release of infectious progeny virions and release of alarmin/DAMP including high mobility group box-1 (HMGB1), IL-18, and IL-16 [60]. Another study led by Daugherty et al. identified that caspase recruitment domain family member 8 (CARD8) could be cleaved by SARS-CoV-2 3CLpro and activates the human CARD8 inflammasome via proteolysis within the disordered N-terminus, demonstrating that it could act as an innate immune sensor of infection by positive-sense RNA viruses including SARS-CoV-2 [61]. Therefore, various NLRs could sense multiple SARS-CoV-2 structural and nonstructural proteins, thus limiting virus replication via activation of inflammasomes and playing important roles during the virus–host interaction.

## 4. SARS-CoV-2 and RLRs

Retinoic-acid-inducible gene I (RIG-I)-like receptors (RLRs) are a family of DExD/H box RNA helicases that function as cytoplasmic sensors of PAMPs within viral RNA and thus can detect a broad range of viruses. They could mediate the transcriptional induction of genes encoding type I IFNs and proinflammatory cytokines that elicit an intracellular immune response to control virus infection [62,63]. RLRs encompass three members: RIG-I, melanoma differentiation-associated protein 5 (MDA5), and laboratory of genetics and physiology 2 (LGP2) [64]. Structurally, they all harbor an ATPase containing DEAD box helicase (DEAD helicase) with the capacity to hydrolyze ATP and to bind and possibly unwind RNA. RIG-I and MDA5 each contain two copies of a caspase recruitment domain (CARD) at the N terminus that mediate downstream signaling transduction and the transcriptional activation of interferons and antiviral response genes while LGP2 lacks the CARDs and is widely believed to regulate RIG-I and MDA5 [64,65,66]. RIG-I and LGP2 also harbor a repressor domain (RD) in their C-terminal regulatory domains (CTDs) [67]. Previous studies have revealed that different RLRs sense different spectrums of RNA viruses, which is mostly attributed to the distinct RNA structures produced by different viruses [68]. Upon binding viral RNA and oligomerization, the RLRs recruit multiple copies of their adapter molecule, the mitochondrial membrane-associated MAVS, via association of their shared CARDs. Next, MAVS further aggregates and recruits additional proteins such as TNF-receptor-associated factors (TRAFs), leading to the activation of TBK1/IKKε (TANK-binding kinase 1/ inhibitor of nuclear factor kappa B kinase ε). Then, IRF3 and IRF7 are activated and together with NF-κB induce type I and type III IFN production and the expression of host defense genes [64,69].

RIG-I preferably binds long dsRNA molecules and short dsRNAs bearing a tri- or di-phosphorylated 5′ end, whereas MDA5 recognizes longer (kilobase-scale) genomic viral RNAs and replication intermediates [70,71,72,73]. RIG-I and MDA5 have been reported to sense different RNA viruses, thus also playing key roles in regulating immune response upon SARS-CoV-2 infection. Kouwaki et al. first confirmed in HEK293 cells that SARS-CoV-2 viral RNAs activated RIG-I and MDA5, and further identified that viral RNA region 24,001–24,200 is crucial for recognition by RLRs and downstream interferon expression [74]. Yin et al. demonstrated that in lung epithelial cells, SARS-CoV-2 sensing mainly relies on MDA5, and downstream IRF3, IRF5, and NF-κB are the key transcription factors involved in the IFN response to SARS-CoV-2 infection. Rebendenne and colleagues confirmed the key role of MDA5 while also noticing that the elicited IFN response was unable to control viral replication in lung cells [58,75]. The possible reason might be explained by another study by Thorne et al., in Calu-3 cells, which showed that both RIG-I and MDA5 can sense SARS-CoV-2 and drive the inflammatory responses, including IFN responses and expression of interferon-stimulated genes irrespective of whether virus replication is suppressed, but the immune responses were relatively preceded by SARS-CoV-2 replication [76]. Moreover, Yamada et al. found that in primary human alveolar and bronchial epithelial cells, mRNA levels of IFNs and cytokines were hardly upregulated while viral replication was suppressed. The underlying mechanism revealed that RIG-I could detect viral infection through the interaction of its helicase domain with the 3′ untranslated region (UTR) of positive-strand viral RNA. This interaction further blocked viral RNA-dependent RNA polymerase (RdRp) from accessing genomic RNAs, thus exhibiting an inhibitory effect on viral replication without activating the conventional RIG-I/MAVS pathway [77]. This suggests that RLRs may affect SARS-CoV-2 infection via different mechanisms.

## 5. Other Sensors during SARS-CoV-2 Infection

Beyond the above-mentioned TLRs, NLRs, and RLRs, other sensors could also recognize viral components and elicit immune responses. The cytosolic DNA-sensing pathway known as the cGAS-stimulator of interferon genes (cGAS-STING) signaling can be activated upon cGAS (Cyclic GMP-AMP synthase) recognition of DNA fragments in the cytoplasm, which further activates the enzymatic activity of cGAS to synthesize 2′3′-cGAMP (cyclic guanosine monophosphate-adenosine monophosphate) using ATP and GTP, and cGAMP subsequently activates STING [78,79,80]. Finally, STING recruits and activates the kinases TBK1 and IKKβ resulting in the activation of IRF3 and NF-κB, leading to the activation of IFNs, ISGs, and proinflammatory cytokines, which in turn counteract viral replication [81]. Though cGAS mainly recognizes DNA, accumulating evidence indicates that cGAS-STING activation can also be activated by RNA viruses and in turn inhibit virus replication [82]. SARS-CoV-2 infection could also be indirectly sensed by the cGAS-STING pathway. It has been revealed that cGAS can recognize chromatin DNA shuttled from the nucleus as a result of cell-to-cell fusion mediated by the SARS-CoV-2 spike and ACE2 receptor, thus contributing to interferon and pro-inflammatory gene expression, confirmed in cells either by ectopic spike protein or VSV-spike pseudovirus [83,84]. In addition to chromatin DNA release, SARS-CoV-2 can also activate the cGAS-STING pathway in endothelial cells through disruption of mitochondrial homeostasis, swollen mitochondrial surfaces, and disrupted cristae mitochondrial DNA release, thus leading to activation of cGAS-STING [85].

Li et al. also found that at the cellular level, SARS-CoV-2 infection can activate oligoadenylate synthetases (OASs) and protein kinase R (PKR) that senses dsRNA, and further induce host antiviral pathways independently of MAVS signaling [86]. Moreover, upon binding to viral or endogenous dsRNA, OASs, primarily OAS3, can synthesize 2′-5′-linked oligoadenylates, induce dimerization and activation of RNase L to cleave viral or host ssRNAs at UN^N motifs [87,88], and thus degrade viral RNAs to inhibit virus replication [89]. While recently, Wickenhagen et al. revealed that OAS1, infrequently considered a major viral dsRNA sensor, can sense several regions of the SARS-CoV-2 genome, with stem loops 1 and 2 (SL1 and SL2) within the 5′-UTR present in all SARS-CoV-2 positive-sense viral RNAs being the major target, and instigate antiviral activity through activation of downstream RNase L [90] (Summarized in Figure 2). Besides epithelial cells, T cells also express PRRs including TLRs and Wu et al. discovered that the triggering receptor expressed on myeloid cells 2 (TREM-2) binds SARS-CoV-2 M protein and interacts with the T cell receptor (TCR) subunit CD3ζ and the kinase ζ-chain-associated protein of 70 kDa (ZAP70) in T cells. Using recombinant M protein, they found M and TREM2 protein interaction induced phosphorylation of CD3ζ, ZAP70, and STAT1 (signal transducer and activator of transcription 1), and further facilitated expression of T-box transcription factor TBX21 (T-bet). In addition, TREM-2 also promotes T_H_1 cytokine IFN-gamma and TNF-α production to defend against virus infection [91]. Taken together, numerous novel and unexpected mechanisms could be adopted by different cell types to regulate host immune response and inflammation to combat COVID-19.

## 6. SARS-CoV-2 Antagonism

With the existence of this impressive array of cellular sensors to antagonize virus infection, SARS-CoV-2 viruses have evolved numerous approaches to either modulate or bypass these sensors and downstream signaling pathways. In general, mechanisms employed by SARS-CoV-2 to modulate host innate immune host defense could be divided into: (1) host key protein cleavage by viral proteases; (2) host translation shut-off; (3) deISGylation of host proteins; (4) reduction in host protein phosphorylation; and (5) prevention of transcription factor translocation [92].

SARS-CoV-2 structural proteins have been reported to be involved in inhibiting and antagonizing innate immune response. SARS-CoV-2 M protein, the structural protein essential in viral assembly, can inhibit innate immune response via directly interacting with RIG-I, MAVS, and TBK1 thus preventing the RIG-I-MAVS, MAVS-TBK1, and TRAF3-TBK1 interactions, MAVS aggregation and downstream recruitment of downstream components, and suppression of the phosphorylation of IRF3, TBK1, IKKα/β, and p65, to attenuate innate immune response and enhance viral replication [93,94]. The N protein of SARS-CoV-2 has multiple roles in dampening innate immune response by: (1) inhibiting the phosphorylation of STAT1 and STAT2 by direct interaction [95]; (2) suppressing IFN via interference with the interaction between tripartite motif-containing protein 25 (TRIM25) and RIG-I, and also the interaction of TBK1 and IRF3 [96]; (3) undergoing liquid–liquid phase separation with RNA, which contributes to virion assembly, viral transcription and replication, inhibition of Lys63-linked poly-ubiquitination, and aggregation of MAVS [97]; (4) targeting host Ras GTPase-activating protein-binding protein 1 (G3BP1) and PACT, upstream cofactors of RLRs signaling pathway to inhibit IFN expression by suppressing the formation of stress granules; and (5) wrapping the genomic RNA to affect the recognition of dsRNA by RIG-I [98]. Interestingly, it has also been proposed by Zhao et al. that the dual role of the N protein in regulating immune response as a low dose suppresses the induction of interferon (IFN-I) signaling and inflammatory cytokines, while high doses in turn had a promotional role [99]. Moreover, SARS-CoV-2 spike protein was reported to interact with IRF3 and mediates its proteasomal degradation thus evading the host innate antiviral immune response [100]. These facts suggested that SARS-CoV-2 structural proteins play pleiotropic roles in regulating and antagonizing host innate immune response.

SARS-CoV-2 encodes 2 proteases, nsp3 and nsp5, that are required to cleave polyprotein into 16 nonstructural proteins (nsp1–nsp16) (Figure 1). They also participate in antagonizing host immune response either by directly targeting host proteins or deubiquitinating and de-ISGylating activities. Nsp5, or the main protease (Mpro), was reported to directly bind and cleave RIG-I at the Q10 residue of the N terminus and abolish its activity to activate MAVS to inhibit downstream IFN signaling. Moreover, nsp5 also targets the K136 residue of MAVS for ubiquitination to promote protein degradation via the ubiquitin–proteasome system [101]. Nsp5 was also reported to cleave NF-κB essential modulator NEMO to deregulate immune response [102,103]. Ruin et al. reported that nsp5 is also a potent inhibitor against the cGAS/STING pathway, its protease activity is important but the underlying mechanism is still unclear [104]. Another important viral protease nsp3 has been reported to attenuate type I interferon responses via cleavage of ISG15 from IRF3 [105]. PLpro could also directly cleave IRF3 to abolish downstream responses [106]. Besides protease activity, PLpro also possesses deubiquitinating and de-ISGylating activities [105,107]. ISGylation of MDA5 is essential for oligomerization, and de-ISGylation by nsp3 abolishes downstream antiviral immune response. Additionally, ISG15 cleavage by nsp3- and nsp3-mediated de-ISGlyation of IRF3 also attenuates immune responses [107,108].

Beyond structural proteins, non-structural and accessary proteins also have important roles in antagonizing host innate immune response. It has been reported that nsp1, nsp6, and nsp12–nsp15 can all suppress innate immune responses with various mechanisms [109]. Nsp1, nsp3, nsp12, nsp13, nsp14, ORF3, ORF6, and M protein can all inhibit Sendai-virus-induced IFN-β promoter activation [110]. ORF9b, ORF3, ORF6, ORF7a, and ORF7b can suppress IFN-I responses to different extents [111]. Nsp1 and nsp14 (with cofactor nsp10) can inhibit translation and thus subsequently block host immune functions including type-I interferon immune response [112,113]. ORF9b can inhibit RIG-I-MAVS antiviral signaling by: (1) interrupting K63-linked ubiquitination of NEMO; (2) interacting with RIG-I, MDA5, and MAVS, perturbing TBK1 phosphorylation; and (3) interacting with translocase of outer membrane 70 (TOM70) to prevent the interaction between TOM70 and heat shock protein 90 (Hsp90) [114,115,116,117]. Nsp6 and nsp13 bind to TBK1 to suppress IRF3 and TBK1 phosphorylation, respectively, and ORF6 was reported to be able to alter IRF3 nuclear translocation to suppress ISRE-dependent gene expression in response to recombinant IFN-I [118]. Since the Jak-STAT pathway is one of the important signaling pathways downstream of cytokine receptors, multiple SARS-CoV-2 proteins affect STAT phosphorylation, which is required for the formation of heterodimer and is associated with IRF9 in forming IFN-stimulated growth factor 3 (ISGF3) to bind interferon-stimulated response elements (ISREs) to transcribe interferon-stimulating genes [119]. Several studies have confirmed that nsp1, nsp6, nsp13, ORF3a, and ORF7b suppressed STAT1 phosphorylation, while nsp6, nsp13, ORF7a, and ORF7b suppress STAT2 phosphorylation [118]. ORF3b, ORF6, ORF8, and nsp13-15 were reported to decrease the nuclear translocation of IRF3 thus inhibiting the production of IFN-I [120,121,122]; ORF10 interacts with NIX to induce mitophagy and degrade MAVS to suppress the downstream innate immune response [123] (Summarized in Figure 3). Importantly, with the emergence of novel variants of SARS-CoV-2, the innate immune suppression effect might change as they may harbor mutations that confer stronger innate immune suppression through enhanced expression of specific viral antagonist protein, enhancing replication through reducing or delaying early host innate responses that may contribute to virus transmission advantage [124]. Or otherwise, mutations that attenuate viral innate immune response may cause replication defects due to elevated immune response, as exemplified by mutation in ORF7a leading to truncation [125], suggesting the continual monitoring of variants and revealing their functions are essential for the better understanding of virus properties. All the interactions between viral and host proteins suggest that elucidating the functions of SARS-CoV-2 proteins in regulating cellular innate immune response is important for a better understanding of viral evolution, transmission, and would shed light on novel antiviral therapies.

## 7. Potential Use of Protein Adjuvant and Therapeutics

Since its first emergence, huge efforts have been put into developing antiviral therapies against SARS-CoV-2 and scientists have focused mainly on virus proteins, especially nsp3, nsp5, and RdRp [126,127,128]. The innate immune response has undoubtfully helped combat virus infections. On the other hand, the interaction between viral PAMPs and PRRs in immune cells may be simultaneously responsible for tissue injury associated with severe virus-induced inflammation [31]. Induction of a well-balanced innate immune response is central to controlling COVID-19 [129,130,131,132]. Therefore, these host sensors can also be harnessed and targeted to a balanced immune response and could be used as vaccine adjuvants for better immune response and therapy or to prevent and control the cytokine storm.

Vaccines played an important role in conferring protection against pathogens; therefore, high-efficacy vaccines against SARS-CoV-2 are required for better control of the pandemic, and proper adjuvants to COVID-19 vaccines can substantially reduce the number of required doses and improve efficacy or cross-neutralizing protection for both subunit vaccines and certain inactivated vaccines [133]. Studies have proved that TLR receptor agonists have elicited robust immune responses. Kurup et al. reported an inactivated rabies-vectored SARS-CoV-2 S1 vaccine CORAVAX, which adjuvanted with TRL4 agonist MPLA-AddaVax elicited high levels of neutralizing antibodies and conferred protection in a Syrian hamster model [134]. Another group led by Chun and colleagues showed that RBD and S1 subunit vaccine admixed with TLR1/2 and TLR3 agonists L-Pampo induces robust humoral and cellular immune responses in a ferret model [135]. Another RBD subunit vaccine was conjugated with the TLR1/2 agonist Pam3CSK4 and the supplement of Pam3CSK4 significantly enhanced the anti-RBD antibody response and cellular response [136]. An amphiphilic imidazoquinoline (IMDQ-PEG-CHOL) TLR7/8 adjuvant formulated with trimeric spike protein induced higher microneutralization titers in a mouse model compared with commonly used adjuvant AddaVax [137]. In addition, 3M-052, an emerging toll-like receptor 7/8 (TLR-7/8) agonist formulated with alum adjuvant RBD vaccine showed better performance in non-human primates compared with alum adjuvant RBD [138]. Additionally, a few STING agonist-based adjuvants such as CF501 showed promising results in stimulating innate immune response when boosting an RBD-Fc vaccine in mice, rabbits, and NHPs [139]. Several other TLR agonist-based adjuvants are currently under evaluation and will help boost future vaccines with better efficacy.

Besides the protective role of the innate immune response, it also contributes to the pathogenesis of SARS-CoV-2, as activation of multiple inflammatory pathways may lead to hyperinflammation and cytokine storm, thus resulting in tissue damage, acute respiratory distress syndrome (ARDS), and multi-organ failure [140]. Therefore, targeting PRRs and related pathways could attenuate the hyperactivation of immune response and reduce cytokine storm. Indeed, several studies have demonstrated that PRR antagonists have beneficial roles and might be used as potential therapeutic drugs for COVID-19, though the exact dosage and duration should be carefully evaluated and tested before clinical use [141,142,143]. Timing is crucial for effective treatment against inflammation caused by viral infection [144]. In animal experiments, TLR2 inhibitor oxPAPC protects against viral pathology in mice [15], famotidine inhibited expression of TLR3 and downstream TLR3-dependent signaling processes [145], and aptamers blocking Spike-TLR4 interaction developed by Yang et al. exhibited robust anti-inflammatory potential [146]. There are several potential drugs targeting TLRs under evaluation in clinical trials including resveratrol targeting TLR4, and M5049 targeting TRL7/8 [147]. Besides TLRs, NLRP3 and inflammasome are also involved in cytokine storm in COVID-19 infections as NLRP3 inflammasome facilitates inflammation by producing IL-1β/18 and causing pyroptosis [148]. Thus, inhibition of NLRP3 is also a promising option in treating patients as it has been shown in a mice model that inhibition of the NLRP3 inflammasome attenuated the release of COVID-19-related pro-inflammatory cytokines, and specific inhibition of the NLRP3 inflammasome by MCC950 relieved lung inflammation and pathology in the mice model [149]. Clinical trials of several NLRP3 inflammasome inhibitors are ongoing and one completed trial indicated that the use of colchicine, which is capable of inhibiting NLRP3 inflammasome by preventing caspase activation, reduces the length of supplemental oxygen therapy and hospitalization and increases clinical improvement (TrialTroveID-381747) [148]. With the attenuated virulence of the Omicron variant and even forms endemic when the pandemic is over, people may treat infections without going to hospitals, thus more FDA-approved and targeted therapies are required to relieve inflammation and hyperactivation of immune responses. Therefore, more understanding is required for virus–host interactions with regard to the innate immune response.

## Figures and Tables

**Figure 1 viruses-15-00352-f001:**
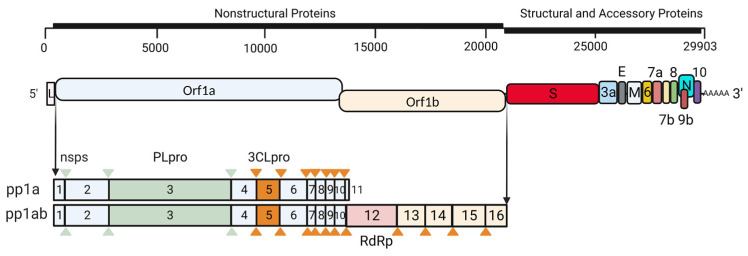
Schematic presentation of SARS-CoV-2 genome and encoded proteins. SARS-CoV-2 encodes 29 proteins including 4 structural proteins, 16 non-structural proteins (nsps), and 9 accessory proteins. Polyprotein pp1a and pp1ab are cleaved by PLpro or 3CLpro as indicated by triangles with different colors to generate 16 nsps.

**Figure 2 viruses-15-00352-f002:**
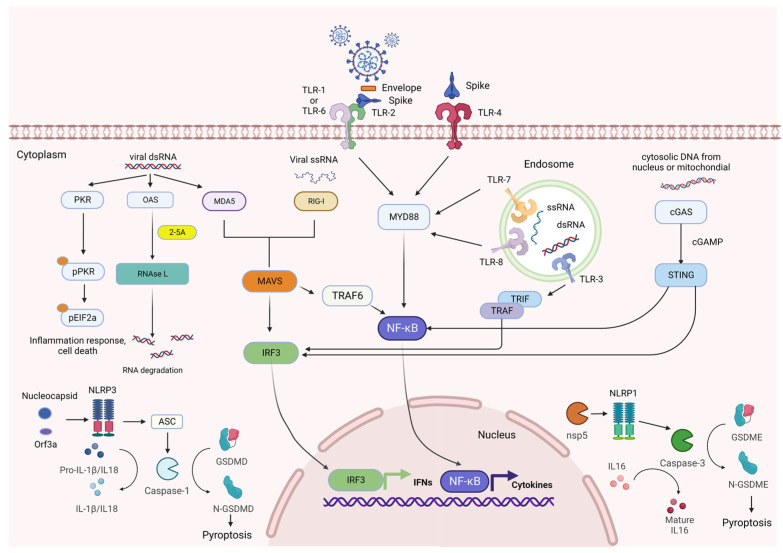
SARS-CoV-2 viral components could be sensed by the innate immune system via PRRs including toll-like receptors, RIG-I-like receptors, NOD-like receptors, and other cellular sensors. Viral RNA components could be sensed by toll-like receptors TLR3, TLR7, TLR8; RLR RIG-I, and MDA5, together with OAS and PKR. SARS-CoV-2 proteins could be detected by TLR2, TLR4, NLRP3, and NLRP1. SARS-CoV-2 infection leading to the release of DNA from the nucleus and mitochondrial could be detected by cGAS and initiate downstream responses.

**Figure 3 viruses-15-00352-f003:**
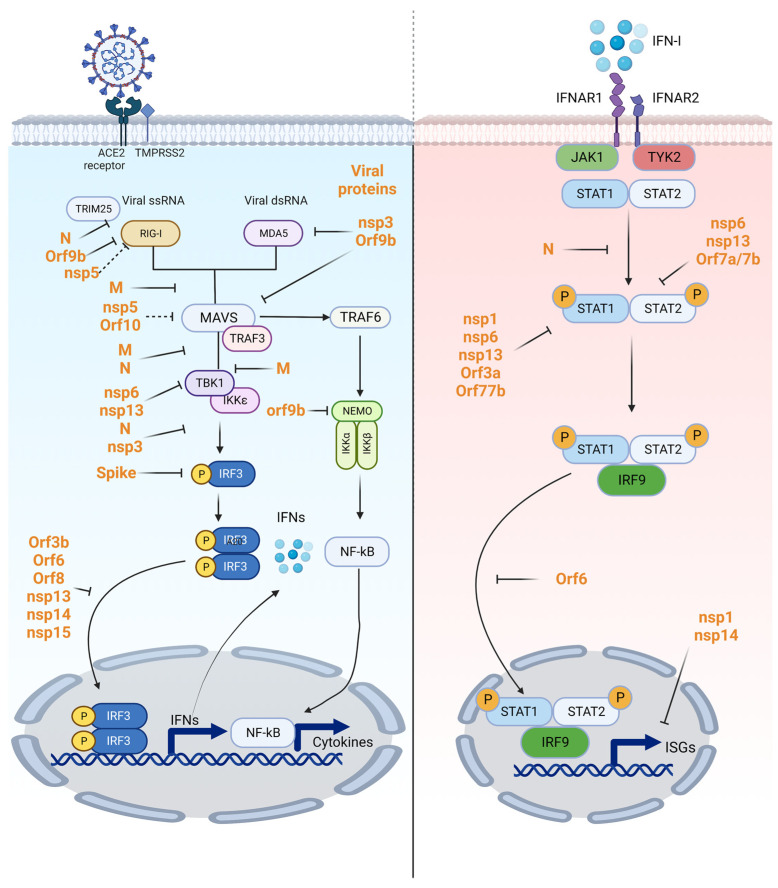
The role of SARS-CoV-2 proteins during antagonizing PRRs and innate immune response. Viral component recognition initiates a signaling cascade culminating in primarily type I interferon (IFN-α/β) and inflammatory cytokine production. IFNs are secreted cytokines that activate a signal transduction cascade leading to the induction of interferon-stimulated genes (ISGs) to restrict virus replication. SARS-CoV-2 proteins can target different host targets to evade immune response at different stages.

## Data Availability

All data used and/or analyzed during this study are included in this published article.

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
