# Peer review of "Cellular Sensors and Viral Countermeasures: A Molecular Arms Race between Host and SARS-CoV-2"

_viruses, 2023, doi:10.3390/v15020352_

Round 1

Reviewer 1 Report

The inaccurate expressions, and mistakes in the letter case and Greek letters exist throughout this article, and grammatical errors stand out as well. Frankly speaking, I spend a hard time reading this article and summarizing the revision suggestions. Please authors carefully check everything in the article before submission next time. The article mainly describes the host cellular sensors, their interactions with viral components, and subsequent effects on viral replication. At the end, authors discussed the potential of host cellular sensor ligands to develop vaccine adjuvants and therapeutics against immunopathology. The title is more interesting. I would advise including more discussions on the fierce 'molecular arms race' of host cellular sensors against SARS-CoV-2 virus, maybe describing the evolution course or SNP diversity of host components and their altered functions during SARS-CoV-2 replication. Emphasizing and reflecting the horror balance in the arms race sounds more interesting. This is just a suggestion for thinking.

Some sentences read very confusing. What's the meaning of 'being a membrane' in line 15? This sentence could be expressed in this way for your reference 'SARS-CoV-2 belongs to the subgenus of sarbecovirus from coronaviridae family, and bears a positive-sense single-stranded RNA genome.'

In my impression, certain protein antigens from SARS-CoV-2 can also bind to NLR family proteins such as NLRP3. A suggestion for your reference is to mention that protein antigens can also be sensed by PRR in the Abstract.

These innate immune pattern recognition molecules are essential for controlling inflammation through the induction of cytokines, chemokines, and anti-microbial genes, which may participate in regulating rather than triggering adaptive immune responses. I think recognition of antigens by lymphocyte receptors can be regarded as triggering.

Suggest deleting 'with' at line 26.

At line 29, with an RNA genome size of approximately 29.9 kb,... recombinations...

At line 30, it is better to use ', which' than 'that'. I would refuse to use the word 'novel viruses'. 'altered characteristics' is not clear, the virus antigenicity, the structure of viral genome, protein expression, or something else?

At line 31, 'existing specific immunities...'

At line 37, '... replication machines of host cells to support...'. I am not sure if it is right saying 'host functions' herein.

At line 39, 'are able to control the infection'.

At line 43, do you agree that the mucosal layer is the first defense line?

At line 43, 'rapidly drive innate cellular responses'.

At line 46, SARS-CoV-2.

At line 50, Upon viral attachment to host cells, the viruses fuse to the cellular membrane and release the viral genome to initiate the virus replication cycle.

I cannot fully understand the sentence in lines 58 - 60.

Could you please give examples of these cytokines that activate adaptive immune responses?  

Suggest deleting various at line 62.

At line 67, it is better to use 'relieve'.

At line 72, suggest adding a conjunction in the sentence.

At line 85, it is suggested to use IFN-γ instead of IFN-G. Or authors should explain IFN-G. Likewise, IL-1β instead of IL1B at line 86, and TLR2 instead of Tlr2 at line 88. IFN-B at line 81. Please carefully check this kind of mistake throughout the article.

At line 113, had been reported instead of had been done.

At line 115, SARS-CoV-2 instead of SARS-COV-2 to keep the virus name consistent.

At line 119, the full name of ssRNA should be explained at its first appearance in the article, which should be line 111. Authors should carefully check the first appearances of many other abbreviations in this article.

What is SAMPs at line 130?

Could you please further explain and give examples of the accumulation of proinflammatory and cytotoxic mediators, the types of infiltrated immune cells in the lungs, or where else?

The last paragraph of section 2 has a very long sentence, it reads quite confusing to me. It is suggested to add the full stop behind 'dsRNA [34, 35]' at line 138, 'of autoimmune disorders' at line 139, 'and immune pathogenesis' at line 145. Then rewrite the sentence in lines 140 - 142 and the sentence in lines 145 and 146.

Suggest using 'response' instead of 'immunity' at line 143.

I guess, 'closely related' at line 157 means that NLRP3 can sense a variety of DAMPs and PAMPs during infection and cellular stress. I still cannot understand the reasons for the expression 'closely related to DAMP and PAMP', could you explain the special features or aspects of NLRP3? The sentence spanning lines 156 - 160 is quite long, it is suggested to revise using shorter sentences and make it more clear.

Suggest revising the sentence at line 175 '... drive pyroptosis that potentially contributes to the COVID-19 pathology'.

Suggest revising the sentence at line 182 '... by caspase 1, which leads to the lytic inflammatory cell death and the pyroptosis.'

Suggest revising the sentence at lines 188 -192. These sentences are hard to read.

Use the Greek letter in NF-κB at line 194.

Suggest using superscript K+.

Use 'lead' instead of 'leading' at line 208.

Suggest revising the sentence at lines 209 - 212. This sentence is hard to read.

NSP5 is used at line 213, while lowercase letter nsp were used in other places, what's the difference?

Revise the sentence at lines 264 - 272. This sentence is too long and contains grammatic error.

'Despite' in line 316 sounds weird in this sentence. This clause gives the first impression that viruses won't be sensed or harmed at all. Suggest to make this correction, '..., SARS-CoV-2 viruses have evolved numerous approaches to either modulate ...'.

What is a serious of studies in line 318? Could you also please revise this sentence in lines 318 - 323 clearer?

inhibiting at line 331.

Suggest correcting the title of section 7. This title is suggested for your reference 'Potential use of protein adjuvant and therapeutics'.

Author Response

The inaccurate expressions, and mistakes in the letter case and Greek letters exist throughout this article, and grammatical errors stand out as well. Frankly speaking, I spend a hard time reading this article and summarizing the revision suggestions. Please authors carefully check everything in the article before submission next time. The article mainly describes the host cellular sensors, their interactions with viral components, and subsequent effects on viral replication. At the end, authors discussed the potential of host cellular sensor ligands to develop vaccine adjuvants and therapeutics against immunopathology. The title is more interesting. I would advise including more discussions on the fierce 'molecular arms race' of host cellular sensors against SARS-CoV-2 virus, maybe describing the evolution course or SNP diversity of host components and their altered functions during SARS-CoV-2 replication. Emphasizing and reflecting the horror balance in the arms race sounds more interesting. This is just a suggestion for thinking.

Some sentences read very confusing. What's the meaning of 'being a membrane' in line 15? This sentence could be expressed in this way for your reference 'SARS-CoV-2 belongs to the subgenus of sarbecovirus from coronaviridae family, and bears a positive-sense single-stranded RNA genome.'

Response: Thank you for the suggestions, this sentence has been revised accordingly.

In my impression, certain protein antigens from SARS-CoV-2 can also bind to NLR family proteins such as NLRP3. A suggestion for your reference is to mention that protein antigens can also be sensed by PRR in the Abstract.

Response: Thank you for the suggestions. This has been added in the abstract as “SARS-CoV-2 belongs to the subgenus of sarbecovirus from coronaviridae family, and bears a positive-sense single-stranded RNA genome. It mainly replicates in cytoplasm and viral components including RNAs and proteins could be sensed by pattern recognition receptors including toll-like receptors (TLRs), RIG-I like receptors (RLRs), Nod-like receptors (NLRs) thus regulating host innate and adaptive immune response.”

These innate immune pattern recognition molecules are essential for controlling inflammation through the induction of cytokines, chemokines, and anti-microbial genes, which may participate in regulating rather than triggering adaptive immune responses. I think recognition of antigens by lymphocyte receptors can be regarded as triggering.

Response: Thank you for the suggestions. This has been corrected in the abstract as below = “thus regulating host innate and adaptive immune response.”

Suggest deleting 'with' at line 26.

Response: This has been corrected as suggested.

At line 29, with an RNA genome size of approximately 29.9 kb,... recombinations...

Response: This has been corrected accordingly.

At line 30, it is better to use ', which' than 'that'. I would refuse to use the word 'novel viruses'. 'altered characteristics' is not clear, the virus antigenicity, the structure of viral genome, protein expression, or something else?

Response: Thank you for this suggestion. We have revised the setences as belew “mutations and recombinations occur from time to time which could give birth to novel viruses with altered properties including replication and transmission efficiency, or with altered antigenicity that could evade from existing specific immunities elicited either by vaccination or infection”.

At line 31, 'existing specific immunities...'

Response: “Specific” had been added as suggested.

At line 37, '... replication machines of host cells to support...'. I am not sure if it is right saying 'host functions' herein.

Response: Thank you for the comments. The sentence has been corrected as “After attachment and entry into susceptible cells, they rely mostly on replication machines of host cells to support viral replication and generation of progeny virions.”

At line 39, 'are able to control the infection'.

Response: This has been corrected as suggested.

At line 43, do you agree that the mucosal layer is the first defense line?

Response: Thank you for pointing out this mistake, we agree with the reviewer and have removed the description “is the first defense line that”.

At line 43, 'rapidly drive innate cellular responses'.

Response: This has been corrected as suggested.

At line 46, SARS-CoV-2.

Response: This typo has been corrected.

At line 50, Upon viral attachment to host cells, the viruses fuse to the cellular membrane and release the viral genome to initiate the virus replication cycle.

Response: This has been corrected as suggested.

I cannot fully understand the sentence in lines 58 - 60.

Could you please give examples of these cytokines that activate adaptive immune responses?  

Response: This sentence has been revised to “These receptors mediate pathogen recognition, and then lead to intracellular signaling and subsequently the synthesis of various cytokines. These cytokines then recruit other immune cells, regulate adaptive immune responses, and inhibit viral spreading.”

Suggest deleting various at line 62.

Response: “various” has been deleted as suggested.

At line 67, it is better to use 'relieve'.

Response: As suggested, “relief” has been corrected.

At line 72, suggest adding a conjunction in the sentence.

Response: As suggested, the sentence has been revised to “Furthermore, TLRs are type I integral membrane proteins, and there are 12 members in mice and 10 members in humans,…”

At line 85, it is suggested to use IFN-γ instead of IFN-G. Or authors should explain IFN-G. Likewise, IL-1β instead of IL1B at line 86, and TLR2 instead of Tlr2 at line 88. IFN-B at line 81. Please carefully check this kind of mistake throughout the article.

Response: Thank you for raising this issue, all related contents has been corrected as suggested and carefully checked throughout the manuscript.

At line 113, had been reported instead of had been done.

Response: “had been done” has been corrected to “had been reported”.

At line 115, SARS-CoV-2 instead of SARS-COV-2 to keep the virus name consistent.

Response: Thank you for pointing out this mistake. This issue had been corrected and checked throughout the manuscript.

At line 119, the full name of ssRNA should be explained at its first appearance in the article, which should be line 111. Authors should carefully check the first appearances of many other abbreviations in this article.

Response: Thank you for raising this question, this issue had been corrected and checked throughout the manuscript.

What is SAMPs at line 130?

Response: SAMPs means SARS-CoV-2-associated molecular patterns (SAMPs) and is added in the manuscript.

Could you please further explain and give examples of the accumulation of proinflammatory and cytotoxic mediators, the types of infiltrated immune cells in the lungs, or where else?

Response: Proinflammatory cytokines include TNF-α, IL-1β, IL-6, IFN-α and IFN-γ, cytotoxic mediators include granzyme B and TRAIL, may reflect the recruitment of NK cells to the lungs. The types of infiltrated immune cells in the lungs were not characterized in the cited research, while CD45 and MHC-II level increase was detected in the lung. Infiltration was revealed by lung histology: “Lung histology revealed a marked infiltration of inflammatory cells into peribronchial and perivascular connective tissue and alveolar septal thickening in SAMP-treated mice.” From reference doi: 10.1172/jci.insight.150542

The last paragraph of section 2 has a very long sentence, it reads quite confusing to me. It is suggested to add the full stop behind 'dsRNA [34, 35]' at line 138, 'of autoimmune disorders' at line 139, 'and immune pathogenesis' at line 145. Then rewrite the sentence in lines 140 - 142 and the sentence in lines 145 and 146.

Response: This long sentence had been revised as suggested, “Further investigation by Croci et al. showed that L412F inhibited autophagy, increased frequency of autoimmune disorders. Besides, co-morbidity was found in L412F COVID-19 males with specific class II HLA haplotypes prone to autoantigen presentation, indicating that TLR3 L412F is a severity marker in COVID-19 infection [36]. Additionally, inborn errors of TLR3- and IRF7-dependent type I IFN immunity can underlie life-threatening COVID-19 pneumonia in patients with no prior severe infection [37], further demonstrating the important role of TLRs in response and immune pathogenesis. Therefore, future investigations of key variants in TLRs are required for a better understanding of human genetic determinants of critical COVID-19 pneumonia.”

Suggest using 'response' instead of 'immunity' at line 143.

Response: This had been revised accordingly.

I guess, 'closely related' at line 157 means that NLRP3 can sense a variety of DAMPs and PAMPs during infection and cellular stress. I still cannot understand the reasons for the expression 'closely related to DAMP and PAMP', could you explain the special features or aspects of NLRP3? The sentence spanning lines 156 - 160 is quite long, it is suggested to revise using shorter sentences and make it more clear.

Response: Thank you for raising up this question, The NLRP3 inflammasome has been of great interest, as mutations in the NLRP3 gene are associated with several autoinflammatory diseases in which the mutant NLRPs stimulate ongoing excessive caspase-1 activity, IL-1β production, and often debilitating inflammation. Regarding SARS-CoV-2 infection, NLRP3 is also a well-studied NLR. To faciliate readers, we have added below introduction of NLRP3 as below,

” Among them, activation of NRLP3 is closely related to sensing of a variety of DAMPs and PAMPs. NRLP3 activation can further induce the formation of inflammasomes multiprotein complexes, activate the inflammatory caspases, lead to the production of mature IL-1 and IL-18, and pyroptotic cell death [39].”

Suggest revising the sentence at line 175 '... drive pyroptosis that potentially contributes to the COVID-19 pathology'.

Response: This has been corrected accordingly.

Suggest revising the sentence at line 182 '... by caspase 1, which leads to the lytic inflammatory cell death and the pyroptosis.'

Response: This has been corrected accordingly.

Suggest revising the sentence at lines 188 -192. These sentences are hard to read.

Response: This sentence has been revised as “while in human and murine macrophages, E protein enhances NLRP3 inflammasome activation when stimulated with LPS and poly(I:C). These facts suggest that E protein suppresses NLRP3 inflammasome activation during the early stages of infection while in the later stages, it may enhance NLRP3 inflammasome activation.”

Use the Greek letter in NF-κB at line 194.

Response: This has been corrected and checked throughout the manuscript as suggested.

Suggest using superscript K+.

Response: It has been corrected.

Use 'lead' instead of 'leading' at line 208.

Response: It has been corrected.

Suggest revising the sentence at lines 209 - 212. This sentence is hard to read.

Response: This sentence has been rephased “Planès et al. reported that NLRP1 also senses SARS-CoV-2. They demonstrated NLRP1 could be cleaved at Q333 site by SARS-CoV-2 3CLpro, and triggers inflammasome assembly and cell death. However, SARS-CoV-2 can also counteract the inflammasome signaling by directly targeting and inactivating GSDMD downstream of NLRP1, as GSDMD could be cleaved by SARS-CoV-2 nsp5 at Q333 site.”

NSP5 is used at line 213, while lowercase letter nsp were used in other places, what's the difference?

Response: Thank you for pointing out this mistake. We have checked and corrected this issue to keep nsps identical throughout the manuscript.

Revise the sentence at lines 264 - 272. This sentence is too long and contains grammatic error.

Response: This sentence has been rephrased “Moreover, Yamada et al. found that in primary human alveolar and bronchial epithelial cells, mRNA levels of IFNs and cytokines were hardly upregulated while viral replication was suppressed. The underlying mechanism was revealed that RIG-I could detect viral infection through the interaction of its helicase domain with the 3′ untranslated region (UTR) of positive-strand viral RNA, this interaction further blocked viral RNA-dependent RNA polymerase (RdRp) to access genomic RNAs thus exhibiting an inhibitory effect on viral replication, without activating the conventional RIG-I/MAVS pathway [77]. This suggests RLRs may affect SARS-CoV-2 infection via different mechanisms.”

'Despite' in line 316 sounds weird in this sentence. This clause gives the first impression that viruses won't be sensed or harmed at all. Suggest to make this correction, '..., SARS-CoV-2 viruses have evolved numerous approaches to either modulate ...'.

Response: Thank you for the suggestion. We have rewritten the description “With the existence of this impressive array of cellular sensors to antagonize virus infection, SARS-CoV-2 viruses have evolved numerous approaches to either modulate or bypass these sensors and downstream signaling pathways.”

What is a serious of studies in line 318? Could you also please revise this sentence in lines 318 - 323 clearer?

Response: Sorry for the typo of ‘series’ , and the sentence has been revised to “In general, mechanisms employed by SARS-CoV-2 to modulate host innate immune host defense could be divided into (1) host key proteins cleavage by viral proteases, (2) host translation shut-off, (3) deISGylation of host proteins, (4) reduction of host protein phosphorylation, (5) prevention of transcription factor translocation [92].”

inhibiting at line 331.

Response: It has been corrected as suggested.

Suggest correcting the title of section 7. This title is suggested for your reference 'Potential use of protein adjuvant and therapeutics'.

Response: Thank you for this suggestion. We have make the change accordingly.

Reviewer 2 Report

This review summarizes the innate immune responses to SARS-CoV-2 infection. Although I think some of the cited work needs further validation or more comprehensive studies, the authors reported with objective comments. The review provides valuable resources for researchers with interests in investigating the innate immune responses against SARS-CoV-2 or other coronaviruses, or even other types of viruses. Summaries on the viral proteins' antagonizing effects on PRRs and innate immune responses depict a general map of choosing anti-viral drug development. Overall, this is a well-organized review on innate immune responses against the virus. 

There are some minor grammatical or spelling errors (e.g. line 15, "member" instead of "membrane"), I suggest the authors for a carefully proofread. 

Author Response

This review summarizes the innate immune responses to SARS-CoV-2 infection. Although I think some of the cited work needs further validation or more comprehensive studies, the authors reported with objective comments. The review provides valuable resources for researchers with interests in investigating the innate immune responses against SARS-CoV-2 or other coronaviruses, or even other types of viruses. Summaries on the viral proteins' antagonizing effects on PRRs and innate immune responses depict a general map of choosing anti-viral drug development. Overall, this is a well-organized review on innate immune responses against the virus. 

There are some minor grammatical or spelling errors (e.g. line 15, "member" instead of "membrane"), I suggest the authors for a carefully proofread. 

Response: Thank you for pointing out this mistake, the typo has been corrected and the manuscript carefully examined.

Reviewer 3 Report

Very difficult to understand signification of too many abreviations, without explanations ! Too many details relative to different hypoptetical opinions !

So, the reader lose logical thread of idea ! Shorter phrases and more conclusive !

Author Response

Very difficult to understand signification of too many abreviations, without explanations ! Too many details relative to different hypoptetical opinions !

So, the reader lose logical thread of idea ! Shorter phrases and more conclusive !

Response: Thank you for raising up these concerns, we have summarized the non-standard abbreviations to make the manuscript clearer to readers. Please refer to the section before References.

Non-standard Abbreviations

AIM2: Absent in melanoma 2

ASC: Apoptosis-associated speck-like protein containing a caspase recruitment domain

BMDMs: Bone marrow-derived macrophages

CARD: Caspase recruitment domain

CASP8: Caspase recruitment domain family member 8

cGAS: Cyclic GMP-AMP synthase

CLEC5A: C-type Lectin Domain Containing 5A

CTD: C-terminal regulatory domains

DAMP: danger-associated molecular pattern

DAI: DNA-dependent activator of IFN-regulatory factors

dsRNA: double-stranded RNA

G3BP1: Ras GTPase-activating protein-binding protein 1

GSDMD: Gasdermin D

HMGB1: High mobility group box-1

Hsp90: Heat shock protein 90

IFN: Interferon

IKK: Inhibitor of nuclear factor kappa B kinase

IL: Interleukin

IRF: Interferon-regulatory factor

ISGM3: IFN-stimulated growth factor 3

ISRE: Interferon-stimulated response element

LGP2: Laboratory of genetics and physiology 2

LRR: Leucine-rich repeat

MAVS: Mitochondrial antiviral signaling protein

MDA5: Melanoma differentiation-associated protein 5

mDC: myeloid dendritic cell

MyD88: myeloid differentiation factor 88

NF-κB: Nuclear factor kappa-light-chain-enhancer of activated B cells

NLR: NOD-like receptor

NLRP1: NLR Family Pyrin Domain Containing 1

NLRP3: NLR Family Pyrin Domain Containing 3

Nsp: non-structural protein

OAS: Oligoadenylate synthetases

PBMC: Peripheral blood mononuclear cell

pDC: plasmacytoid dendritic cell

PKR: Protein kinase R

PLpro: Papain-Like protease

PAMP: pathogen-associated molecular pattern

PRR: pattern recognition receptor

RD: Repressor domain

RIG-I: Retinoic acid-inducible gene I

ssRNA: single-stranded RNA

SASP: senescence-associated secretory phenotype

SAMPs: SARS-CoV-2-associated molecular patterns

STAT: Signal transducer and activator of transcription

STING: Stimulator of interferon genes

TBK1: TANK-binding kinase 1

TCR: T cell receptor

TIR: Toll/IL-1 receptor

TNF: Tumor necrosis factors

TLR: Toll-like receptor

TOM70: Translocase of outer membrane 70

TRAF: TNF receptor-associated factor

TRIF: TIR domain–containing adaptor-inducing IFN-β factor

TRAIL: Tumor necrosis factor-related apoptosis-inducing ligand

TREM-2: Triggering receptor expressed on myeloid cells-2

TRIM25: Tripartite motif-containing protein 25

ZBP1: DNA-binding protein 1

3CLpro: 3C-like cysteine protease

Round 2

Reviewer 1 Report

In the revised article, at lines 200, 201, and 276, what is the NRLP? 

It is a pity that I do not see further deeper discussion on the dubbed molecular arms race, a kind of rat race in the course of biological evolution. 

Overall, the revised article is fine with me.

Author Response

In the revised article, at lines 200, 201, and 276, what is the NRLP? 

It is a pity that I do not see further deeper discussion on the dubbed molecular arms race, a kind of rat race in the course of biological evolution. 

Overall, the revised article is fine with me.

Response: Thank you for pointing out this typo, it has been corrected throughout the manuscript.